# Highly conserved and *cis*-acting lncRNAs produced from paralogous regions in the center of HOXA and HOXB clusters in the endoderm lineage

Neta Degani[1], Yoav Lubelsky[1], Rotem Ben-Tov Perry[1], Elena Ainbinder[2], Igor Ulitsky[1]*

1 Department of Biological Regulation, Weizmann Institute of Science, Rehovot, Israel, 2 Department of Life Sciences Core Facilites, Weizmann Institute of Science, Rehovot, Israel

* igor.ulitsky@weizmann.ac.il

**Data Availability Statement:** RNA-seq datasets are deposited in GEO database under the accession GSE168444 (reviewer token olepogycpjsrdcp).

## Abstract

Long noncoding RNAs (lncRNAs) have been shown to play important roles in gene regulatory networks acting in early development. There has been rapid turnover of lncRNA loci during vertebrate evolution, with few human lncRNAs conserved beyond mammals. The sequences of these rare deeply conserved lncRNAs are typically not similar to each other. Here, we characterize *HOXA-AS3* and *HOXB-AS3*, lncRNAs produced from the central regions of the HOXA and HOXB clusters. Sequence-similar orthologs of both lncRNAs are found in multiple vertebrate species and there is evident sequence similarity between their promoters, suggesting that the production of these lncRNAs predates the duplication of the HOX clusters at the root of the vertebrate lineage. This conservation extends to similar expression patterns of the two lncRNAs, in particular in cells transiently arising during early development or in the adult colon. Functionally, the RNA products of *HOXA-AS3* and *HOXB-AS3* regulate the expression of their overlapping HOX5–7 genes both in HT-29 cells and during differentiation of human embryonic stem cells. Beyond production of paralogous protein-coding and microRNA genes, the regulatory program in the HOX clusters therefore also relies on paralogous lncRNAs acting in restricted spatial and temporal windows of embryonic development and cell differentiation.

## Author summary

Each of the four Hox clusters in vertebrate genomes encodes up to 11 transcription factors whose activity is extensively regulated spatially and temporally, and which help determine the developmental and adult transcriptome in space and time. These Hox transcription factors belong to 13 homology groups, and Hox clusters also encode various noncoding transcripts, including microRNAs and long noncoding RNAs (lncRNAs). We characterize in detail two lncRNAs, *HOXA-AS3* and *HOXB-AS3*, which are transcribed from matching regions in the HOXA and HOXB clusters, respectively. These lncRNAs are highly

**Funding:** This study was funded by grants from the US-Israel Binational Science Foundation (grant Number 2015171), Minerva Foundation, Israel Science Foundation grant 1242/14 and European Research Council grant lincSAFARI, all to IU. The funders had no role in study design, data collection and analysis, decision to publish, or preparation of the manuscript.

**Competing interests:** The authors have declared that no competing interests exist.

conserved in vertebrate evolution and transcribed antisense to Hox protein-coding genes from groups 5–7. Beyond the matching positions, the promoters of *HOXA-AS3* and *HOXB-AS3* share sequence similarity, their expression patterns are correlated with each other, mostly in the endoderm lineage, and they positively regulate the expression of the Hox protein-coding genes that they overlap. Regulation by lncRNAs thus appears to be an ancestral feature of HOX clusters, likely pre-dating the duplication of the Hox clusters at the root of the vertebrate lineage.

## Introduction

Over the past decade, genome-wide transcriptome analyses revealed a plaetora of noncoding RNAs, that are expressed from a large number of genomic loci. Among those non-coding genes are long noncoding RNAs (lncRNAs), RNA Pol2 products that are longer than 200 nt. Similarly to mRNAs, lncRNAs begin with a 5' cap and end with a poly(A) tail. To date, thousands of lncRNAs have been reported in different vertebrates [1,2], and it is yet unknown how many of them are functional and what is the full extent of their biological diversity. Many lncRNAs display highly restricted expression profiles during development, potentially allowing them to control gene expression in specific cellular contexts [2,3]. Some lncRNAs have been shown to indeed contribute to proper embryonic development [4].

Mouse and human Hox genes are organized in four genomic clusters (HOXA to HOXD) that exhibit a unique mode of transcriptional regulation–temporal and spatial collinearity–the position of the genes along the chromosome roughly corresponds to the time and place of their expression during development. The sequential activation of Hox genes in the primitive streak helps determine the subsequent pattern of expression along the anterior–posterior axis of the embryo [5,6]. Despite the crucial importance of Hox genes during development [7], the molecular pathways that dictate their collinear expression are not fully understood.

Noncoding RNAs are likely to play important roles in Hox gene regulation. For example, Hox clusters encode two conserved miRNAs, miR-10 and miR-196, that target some of the Hox genes and help establish specific regulatory programs in the embryo [8,9]. One of the first lncRNAs that has been studied in detail, *HOTAIR*, is produced from the HOXC cluster and was reported to regulate expression of HOXD genes [10]. Since this seminal discovery, numerous lncRNAs have been implicated as important in the Hox gene regulation [11]. For example, *HOTTIP*, a lncRNA is located at the 5' end of the HOXA cluster, was shown to control activation of 5' HOXA genes in *cis* via cooperation with an MLL histone methyltransferase complex and chromosomal looping that brings it into close proximity with 5' HOXA gene loci [12].

The protein-coding genes in the four vertebrate Hox clusters belong to 13 groups of orthologs that can be traced to ancestral clusters that existed before the two rounds of genome-duplication [13]. The two conserved microRNA families encoded in the Hox clusters, miR-10 and miR-196, are represented in multiple clusters [14]. lncRNAs have been described in each of the four clusters but so far there were no known cases of clear similarity between lncRNAs across clusters. Here, we focus on a pair of lncRNAs that appear to be some of the most conserved lncRNAs produced from the vertebrate Hox clusters–*HOXA-AS3* and *HOXB-AS3*. We provide evidence that it is likely that the production of these lncRNAs precedes the duplication of the ancestral Hox cluster into HOXA and HOXB. Both lncRNAs are expressed predominantly in the embryo, with expression patterns more similar to each other than to nearby protein-coding genes. In the adult, *HOXA-AS3* expression is mostly restricted to tissues of endodermal lineage, and specifically to immature goblet cells and tuft cells. The similar

expression of *HOXA-AS3* and *HOXB-AS3* is likely driven by conserved and shared binding sites for CDX transcription factors in the *HOXA-AS3* and *HOXB-AS3* promoters. Using human cell lines and human embryonic stem cells, we show that perturbation of *HOXA-AS3* and *HOXB-AS3* expression results in corresponding changes in expression of HOX-6 and HOX-7 genes. These results suggest co-ordinated and ancient lncRNAs production from central regions of the Hox clusters that plays important *cis*-acting gene regulatory roles in cells of the endodermal lineage.

## Results

### A pair of conserved lncRNAs in the middle of HOXA and HOXB clusters

The central regions of HOX clusters give rise to a large variety of transcription products that undergo extensive alternative splicing (**S1A Fig**). We first focused on *HOXA-AS3*, the main transcription start site of which lies ~700 nt downstream of the annotated 3' end of *HOXA5* and which is transcribed antisense to *HOXA5* and *HOXA6*, terminating in the single intron of *HOXA7* (**Figs 1A** and **S1A**). The region in the mouse genome that aligns to the *HOXA-AS3* promoter is the promoter of *Hoxaas3* (*2700086A05Rik*), which terminates in the intergenic region between *Hoxa6* and *Hoxa7* (**Fig 1A**). The promoter of *HOXA-AS3* is highly conserved in other vertebrates, but transcripts originating from it are not consistently annotated, likely due to its very restricted expression in adult tissues, as it is expressed predominantly in the embryo (see below). Using available RNA-seq data we could identify orthologs for *HOXA-AS3* in opossum and *X. tropicalis* (**Figs 1A** and **S1**). Transcription of these orthologs, similarly to that of the human *HOXA-AS3*, started ~500 nt downstream of the 3' end of *HOXA5* and ended in the intron of *HOXA7*. *HOXA-AS3* exhibited significant sequence similarity with the orthologs from mouse, opossum, and *X. tropicalis* (BLAST E-value$<10^{-40}$). Notably, homology with the *X. tropicalis* ortholog was restricted to the region overlapping *HOXA7*.

*HOXB-AS3* transcription in human starts ~900 nt downstream of the 3' end of *HOXB5* and terminates in the intergenic region between *HOXB6* and *HOXB7* (**Figs 1B** and **S1A**). Presumably because of its broader expression compared to *HOXA-AS3*, orthologs of *HOXB-AS3* were readily identifiable in more species. In mouse, it is annotated as *Hoxb5os* (*0610040B09Rik*), and we could identify orthologs in opossum, *X. tropicalis*, coelacanth, spotted gar, medaka, and elephant shark (**Figs 1B** and **S1**). *HOXB-AS3* exhibited significant sequence similarity with the orthologs from mouse and opossum (BLAST E-value$<10^{-40}$), but not with more distant species. Comparison of the sequences with LncLOOM [15] identified four motifs conserved in mammals and in *X. tropicalis* but no deeper conservation was detected (**S1 Dataset**). Both *HOXA-AS3* and *HOXB-AS3* show negative PhyloCSF [16] scores throughout the locus (**S2 Fig**), and so it is unlikely that they encode highly conserved proteins. Notably, a primate-specific protein has been recently found to be encoded by *HOXB-AS3* [17] (see Discussion).

The corresponding positions of the two lncRNAs and the high conservation of their presence in other species made us scrutinize and compare the sequences of their promoters. BLAST comparison of the corresponding promoters from HOXA and HOXB clusters found significant homology in representative vertebrate species all the way to the cartilaginous fish elephant shark (E-value 6e-31 in human, 5e-32 in mouse, 1e-21 in Xenopus, 73–80% base identity). Mapping the transcription start sites of *HOXA-AS3* and *HOXB-AS3* transcripts based on RNA-seq data (where available) suggested that the precise position of transcription initiation varies between the clusters and to a lesser extent between the species (**S3A Fig**). Among the highly conserved sequences preserved in both classes, we note a pair of tandem binding sites for the CDX1/2 proteins—CCATAAA and CCATTAAA [18] that appear once on the sense and once on the antisense strand. When considering all the human promoters

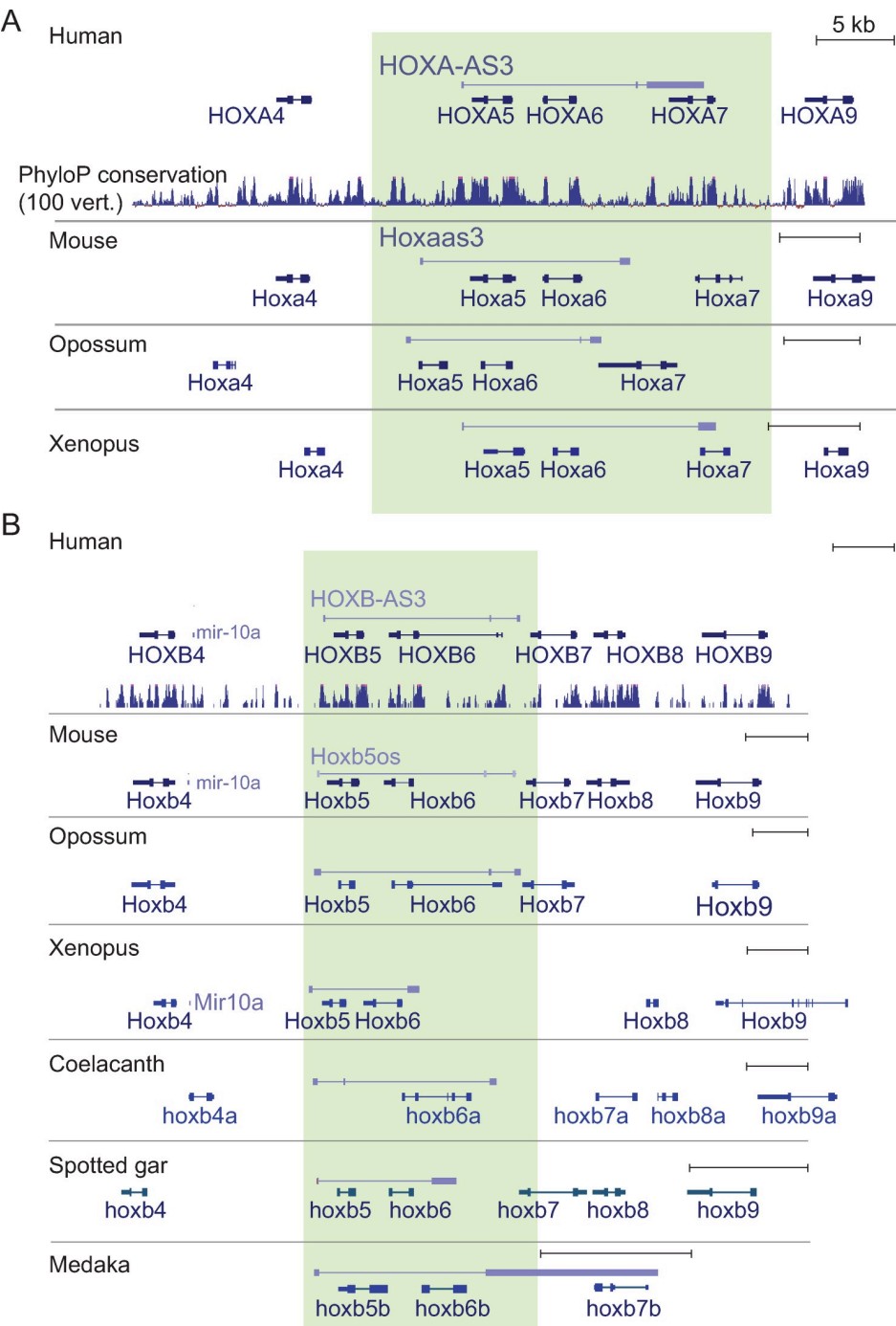

**Fig 1. Orthologs of *HOXA-AS3* and *HOXB-AS3* in different vertebrate species.** Transcript models annotated by Ensembl, Refseq, or PLAR [61], or manually reconstructed based on RNA-seq data (see **S1 Fig**) for *HOXA-AS3* (**A**) and HOXB-AS3 (**B**) are shown alongside the annotated protein-coding genes in the locus. The lncRNAs are transcribed from the '+' strand and all other genes are transcribed from the '-' strand. The regions of *HOXA-AS3* and *HOXB-AS3* are shaded.

annotated in FANTOM5.5 [19], the *HOXA-AS3* promoter contained 11 predicted CDX binding sites, a number of predicted binding sites larger than that in 99.95% of human promoters annotated in FANOM5.5 (only 83 of the 200K promoters had 11 predicted sites or more).

*HOXB-AS3* promoter had two predicted sites, a number comparable to that of several other Hox genes (**S3B Fig**).

## *HOXA-AS3* and *HOXB-AS3* are co-expressed in embryonic and adult tissues

In order to obtain a comprehensive picture of where the two lncRNAs are expressed in both fetal and adult cell types, we relied on data from the FANTOM5.5 project [19], which provide strand-specific data across hundreds of cell types. Both lncRNAs were expressed in a highly specific manner and with patterns largely distinct from those of the overlapping HOX-5, HOX-6, HOX-7 genes (**Figs 2A** and **S4B**). In particular, expression of *HOXA-AS3* in human was more tissue-specific than that of the genes it overlapped and more closely resembled *HOXB-AS3* than any of the genes whose transcription units it overlapped (**Fig 2A**). The correlation between *HOXA-AS3* and *HOXB-AS3* was comparable to the correlations between Hox protein-coding genes with a posterior expression domain (**Fig 2A**), and larger than that found typically between Hox genes in different clusters (**S4A Fig**). In mouse, *Hoxaas3* and *Hoxb5os* were also more tissue specific than the Hox genes they overlapped, but the correlation between them was weaker (**S4B Fig**). Notably, correlations in mouse were more difficult to assess, as *Hoxaas3* was expressed with TPM$\geq$1 only in 17 samples (for comparison, *HOXA-AS3* was expressed with TPM$\geq$1 in 65 human samples). The samples in which *HOXA-AS3* and *HOXB-AS3* were expressed (**S1 Table**) were mostly embryonic or derived from embryonic stem cells. In human, both lncRNAs were co-expressed during late-stage differentiation of embryonic stem cells to embryoid bodies. In mouse, both *Hoxaas3* and *Hoxb5os* were co-expressed in the E8.5 mesoderm in the neonate intestine (consistent with the single cell RNA-seq data, **Fig 2B**).

In order to assess the spatial expression patterns of *Hoxaas3* and *Hoxb5os* and other Hox genes, we reanalyzed Geo-seq data from the mouse E7.5 embryos [20] (**S5 Fig**). Both lncRNAs exhibited specific and overlapping expression domains in the region corresponding to primitive streak or 'late mesoderm' (sections 9–10, posterior region), consistently with the scRNA-seq data. Notably it has been suggested that some of the cells in this region are endoderm cells that egress through the mesoderm late in gastrulation [20–23]. *Hoxaas3* and *Hoxb5os* expression domain was more specific than that of the overlapping Hox genes, and interestingly, overlapped with the expression of Cdx1, and to a lesser extent Cdx2.

We next examined *Hoxaas3* expression in adult mouse tissues in the Tabula Muris scRNA-seq dataset (*Hoxb5os* is not annotated in this dataset). The only cell type where there was appreciable expression were Goblet and epithelial cells from the large intestine (**S4C Fig**), consistent with our more detailed analysis (see below). *HOXA-AS3* and *HOXB-AS3* thus exhibit a very high tissue specificity in the adult tissues, similarly to other lncRNAs [24].

To obtain single-cell resolution on the expression of the two lncRNAs during early embryonic development, we used the large-scale single-cell dataset recently published by the Sanger institute [25], which profiled mouse embryos at E6.5–E8.5 (**Fig 2B**). At these stages, *Hoxaas3* and *Hoxb5os* were generally more highly expressed than the protein-coding genes overlapping their gene bodies. As in the FANTOM data, *Hoxaas3* expression was most similar to the expression of *Hoxb5os* and the two lncRNAs were highly expressed in neuro-mesodermal progenitors (NPM in **Fig 2B**), various mesodermal populations, caudal epiblast, and gut cells.

## *HOXA-AS3* and *HOXB-AS3* regulate their adjacent Hox genes in HT-29 cells

Inspection of ENCODE data suggested *HOXA-AS3* is not well-expressed in commonly used human cell lines, consistently with its overall low expression in adult tissues. *HOXB-AS3* is

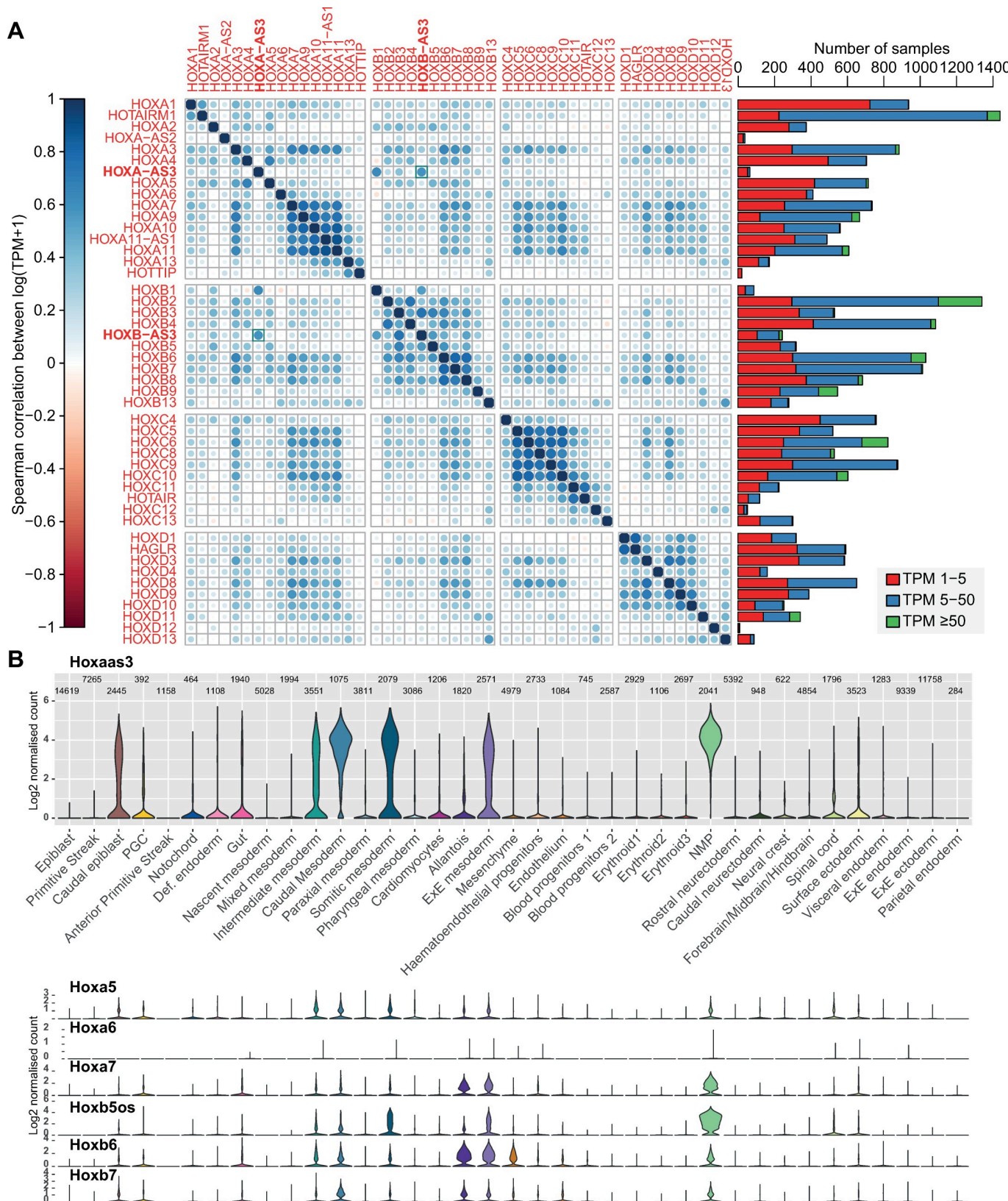

**Fig 2. Expression of HOXA-AS3 and HOXB-AS3 in adult and embryonic tissues. (A)** Left: Correlation coefficients between log-transformed FANTOM5.5 expression levels [19] in hundreds of samples for the indicated genes. Right: Number of samples in which each gene is expressed within the indicated TPM ranges. **(B)** Expression levels of the indicated genes in clusters of single cells during gastrulation, data from [25].

somewhat more broadly expressed, as it is expressed also in Ag04450, IMR-90, and NHLF cells. Surprisingly, there was no substantial expression of *HOXA-AS3* or *HOXB-AS3* in A549 cell line and limited expression in HUVEC, where they have been previously studied [26–27] (**S6 Fig** and Discussion). In contrast, ENCODE RNA-seq data showed that *HOXA-AS3* and *HOXB-AS3* are well expressed in HT-29 (**S6 Fig**)–a human colon adenocarcinoma cell line that under certain growth conditions exhibits characteristics of mature intestinal cells, such as enterocytes or mucus producing cells which have brush borders and expresses Villin and additional intestinal microvilli proteins [28,29].

In order to perturb the expression of *HOXA-AS3* and *HOXB-AS3*, we first used CRISPR interference (CRISPRi) [30]–a catalytically inactive version of Cas9 (dCAS9) optionally fused to a KRAB domain (dCas9-KRAB) together with guide RNAs (gRNAs) directed to a region downstream to the TSS of the target [31]. We transfected HT-29 cells with pools of three gRNAs targeting *HOXA-AS3* and *HOXB-AS3* promoters and dCAS9-KRAB vectors, which reduced lncRNA levels by 50%–80% compared to cells transfected with the dCas9-KRAB vector and an empty gRNA plasmid (**Fig 3A** and **3B**). As *HOXA-AS3* levels are reduced, *HOXA5*, *HOXA6* and *HOXA7* RNA levels are also down-regulated by 30–40% (**Fig 3A**). Similarly, *HOXB-AS3* knockdown (KD) was followed by a down regulation of *HOXB5* and *HOXB6* by 50–70% (**Fig 3B**).

Next, we tested the effect of over-expression (OE) of the lncRNAs using CRISPR activation (CRISPRa)–dCas9 fused to VP64 transcriptional activation domain and directed by the sgRNAs to a region upstream to the *HOXA-AS3* and *HOXB-AS3* TSSs. In this system, the VP64 domain recruits the transcription machinery to activate expression of the lncRNA of interest [31]. OE in HT-29 cells resulted in effects opposite to those observed following lncRNA KD, as it increased expression of the adjacent genes, significantly for *HOXA-AS3* (**Fig 3C and 3D**).

These results suggest that *HOXA-AS3* and *HOXB-AS3* production or their RNA products have a positive regulatory effect on the expression of the neighboring HOX5–7 genes.

## *HOXA-AS3* and HOXB-AS3 RNA products are required for their *cis*-regulatory activity

In order to differentiate between the potential effects on chromatin caused by the use of the KRAB effectors and the transcription or the RNA products of *HOXA-AS3* and *HOXB-AS3*, we used RNAi to target the RNA products of *HOXA-AS3* and *HOXB-AS3*. First we transfected siRNA pools targeting *HOXA-AS3* or *HOXB-AS3* into HT-29 cells. This resulted in a substantial reduction in RNA levels for both *HOXA-AS3* and *HOXB-AS3* and a concomitant reduction in the expression of neighboring genes that was similar to the effects observed with CRISPRi (**Fig 4A** and **4B**). When *HOXA-AS3* was reduced by 60%, *HOXA5/6/7* were significantly downregulated by 20–45% (**Fig 4A**). Similarly, when *HOXB-AS3* was reduced by ~40%, there was a significant downregulation of *HOXB5* and *HOXB6* (**Fig 4B**). As an alternative approach, a stably expressed shRNA targeting *HOXA-AS3* introduced via a lentiviral infection led to a stronger effect with the same trend as that observed using CRISPRi and siRNA, where KD of the lncRNA was accompanied by a decrease of expression of the neighboring genes (**Fig 4C**). *HOXA-AS3* and *HOXB-AS3* RNA products are therefore important for regulation of the adjacent genes.

In order to characterize more broadly the consequences of down-regulation of *HOXA-AS3* and *HOXB-AS3*, we used RNA-seq to profile transcriptome-wide gene expression in HT-29 cells treated with siRNAs targeting these lncRNAs or with a non-targeting control. RNAi resulted in reduction in expression of the lncRNAs, concomitantly with reduction in the

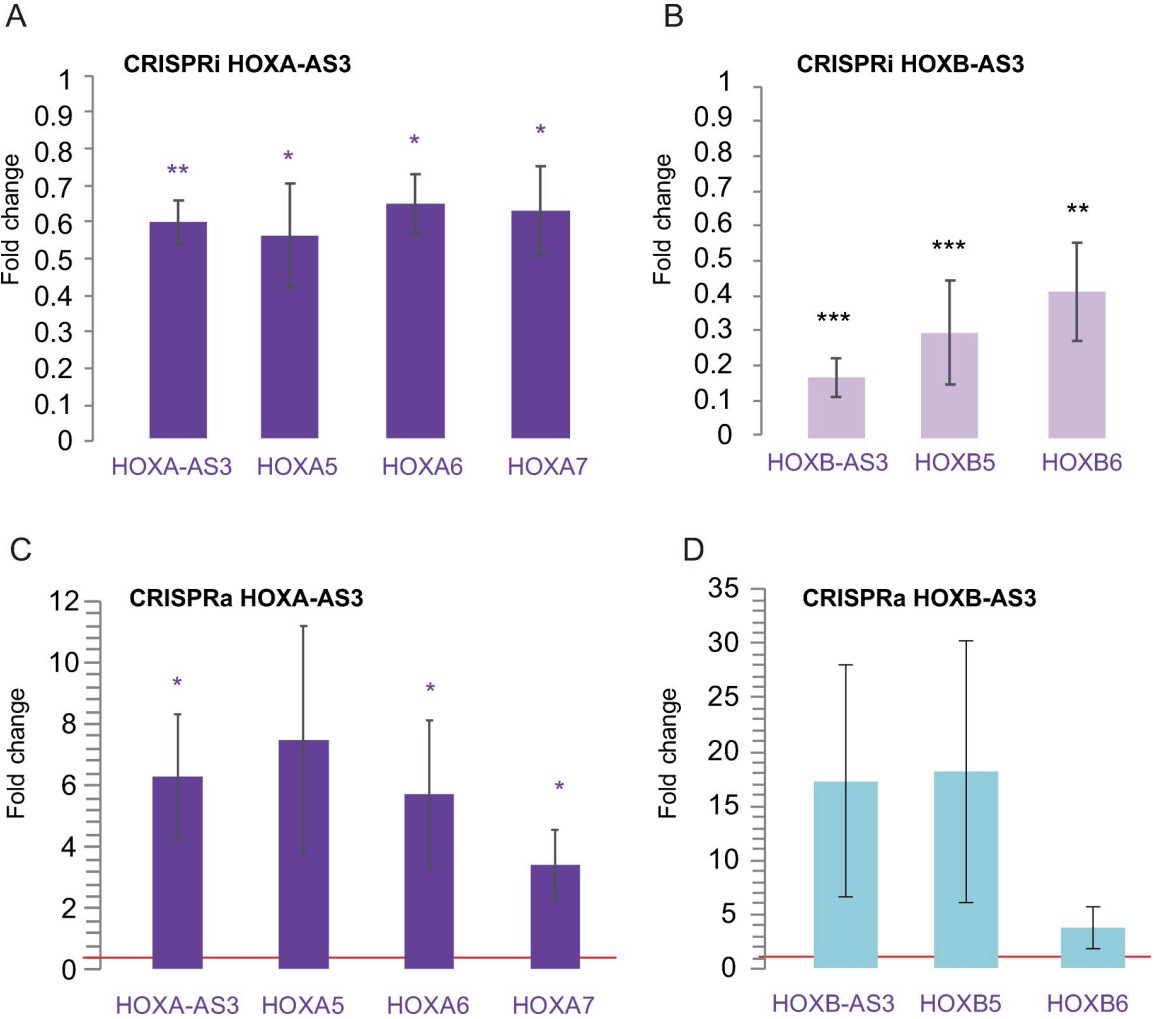

**Fig 3. CRISPR inhibition and activation of *HOXA-AS3* and *HOXB-AS3* in HT-29 cells. (A)** Changes in expression of the indicated genes is shown following inhibition of *HOXA-AS3*. n = 4. (**B**) As in A, following inhibition of *HOXB-AS3*. n = 4. (**C**) As in A, following activation of *HOXA-AS3*. n = 4 (**D**) As in A following activation of *HOXB-AS3*. Normalized to actin. Two-sided t-test. *—P<0.05, **—P<0.005, ***—P<0.0005. Two-sided t-test compared to the transfection control.

overlapping genes, and a broad mild reduction in expression of genes in the HOXA and HOXB clusters (HOXC and HOXD clusters are mostly silent in HT-29 cells) (**Fig 4D** and **S2 Table**), with more significant effects observed in the HOXB cluster that is overall more expressed than HOXA in HT-29 cells (**S7A Fig**). In the case of *HOXB-AS3* it was apparent that the KD had a strong effect on the levels of the overlapping HOXB5–7 genes relative to the other HOX genes. The repressive effect of KD of *HOXA-AS3* on HOXB genes, and of *HOXB-AS3* KD on HOXA genes was validated by qRT-PCR following siRNA KD or CRISPRi of these genes (**S7A** and **S7B Fig**) These results suggest that loss of *HOXA-AS3* and *HOXB-AS3* has broad effects on expression of genes from HOXA and HOXB clusters.

Beyond the effect on the expression of HOX genes, *HOXB-AS3* had a larger effect on gene expression (**S7C Fig**), consistently with its higher expression levels in HT-29 cells. Analysis of the gene expression changes using GOrilla [32] (**S2 Table**) showed that *HOXA-AS3* KD was associated with a significant reduction in genes related to cell cycle and proliferation (top down-regulated GO category "mitotic cell cycle process" adjusted P = 1.52×10$^{-6}$), consistent

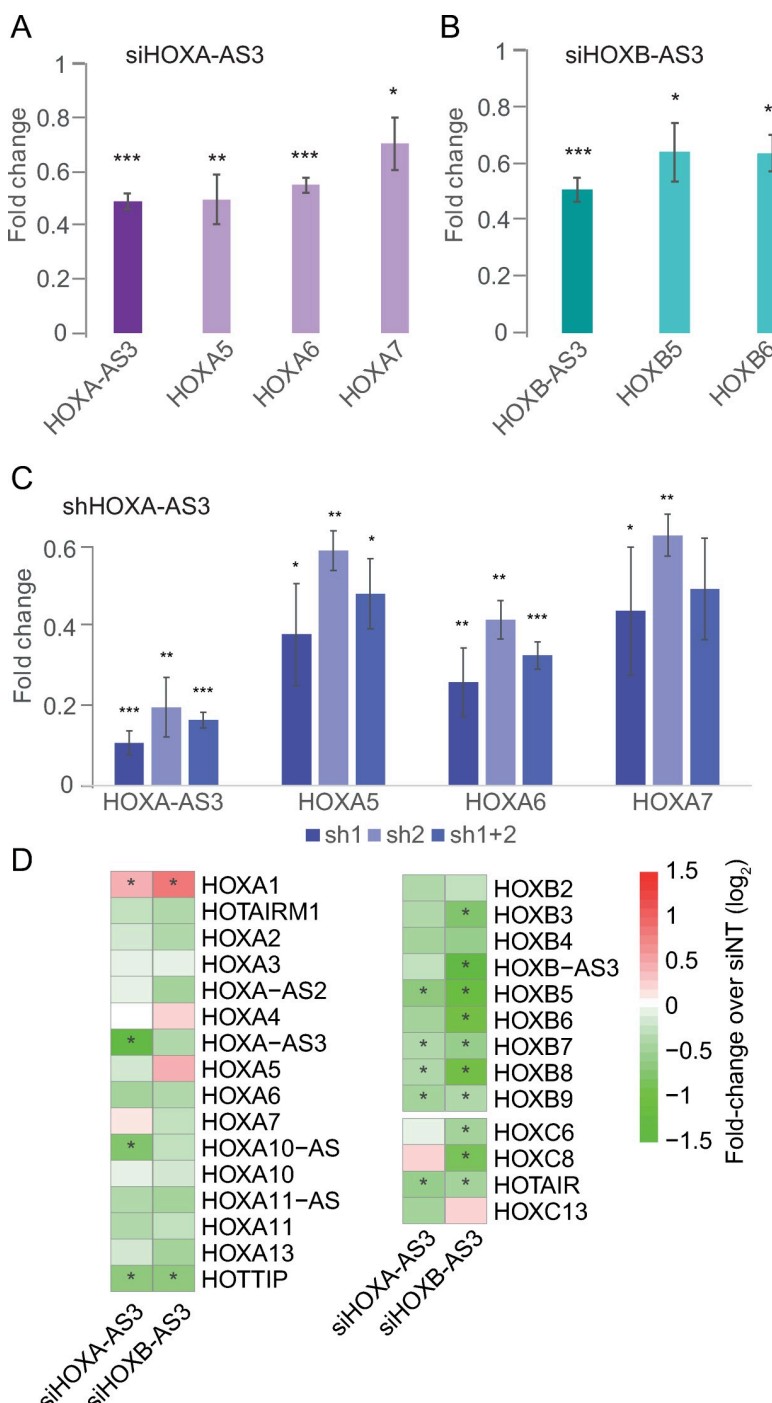

**Fig 4. RNA products of *HOXA-AS3* and *HOXB-AS3* are required for regulation of their adjacent Hox genes. (A-C)** qRT-PCR measurements of the indicated genes in HT-29 cells treated with the indicated reagents. Normalized to Actin. n = 4 for siHOXA-AS3 and siHOXB-AS3. n = 3 for shHOXA-AS3. *—P<0.05, **—P<0.005, ***—P<0.0005. Two-sided t-test compared to the transfection control. **(D)** Changes in gene expression in RNA-seq data of HT-29 cells treated with HOXA-AS3 or HOXB-AS3 siRNAs. Shown are *HOXA-AS3*, *HOXB-AS3*, and all other HOX genes with average FPKM≥1. Asterisks indicate adjusted P<0.05 as computed by DESeq2.

with its reported positive effect of proliferation reported in other cell lines [27,33,34] (see Discussion). *HOXB-AS3* led to a significant up-regulation of genes whose protein products are involved in ncRNA processing, and specifically in rRNA processing (adjusted P = $5.92 \times 10^{-5}$), potentially related to its reported functions in rRNA biogenesis observed in leukemia cells [35]. The changes in gene expression outside of the HOX clusters following *HOXA-AS3* or *HOXB-AS3* KD could result from the consequences of changes in gene expression or from additional trans-acting functions of these lncRNAs (see Discussion).

## *HOXA-AS3* is localized in the both the nucleus and cytoplasm of HT-29 cells

We next focused on *HOXA-AS3* and characterized its precise expression pattern at higher resolution, as it is more narrowly expressed compared to *HOXB-AS3*, and also has a longer exonic sequence which permits the use of Stellaris smFISH protocol with 96 exonic probes for the human *HOXA-AS3* and 94 for the mouse *Hoxaas3* (S3 Table), whereas only 24 probes were possible for *HOXB-AS3*.We first analyzed the subcellular localization of *HOXA-AS3* and *HOXA5* in HT-29 cells (Fig 5A). We observed variable expression of both genes among cells, in some of the cells we could detect expression of only one of the transcripts, while others expressed both genes. *HOXA-AS3* transcript was detectable in just ~15% of the >100 imaged cells, in up to 3 foci per cell and with localization mainly in the nucleus, though it could also be detected in the cytoplasm. Interestingly, in some of the cells that express both *HOXA-AS3* and *HOXA5* we detected a rare yet highly specific co-localization in the perinuclear area (Fig 5B). As expected from their genomic co-location, *HOXA-AS3* and *HOXA5* are co-localized in what is likely their site of transcription in the nucleus (Fig 5B).

## *HOXA-AS3* is expressed in a specific subset of colon epithelial cells

As HT-29 cells contain a mixture of cellular states from the colon epithelium [28,29], *HOXA-AS3* expression in a small subset of cells may imply that it is only found in a defined subpopulation of cells. We therefore analyzed the expression pattern of *HOXA-AS3* and *Hoxaas3* in normal intestinal epithelial cells, using single-cell RNA sequencing (scRNA-seq) data.

In scRNA-seq data from the human colon scRNA-seq data, HOXA-AS3 was expressed predominantly in epithelial cells, and within those it was detected specifically in tuft and immature goblet cells, that are deep crypt goblet cells that are part of the stem cell niche [36] (Fig 6A). Similarly, in the mouse small intestine [37] *HOXA-AS3* is mainly expressed in tuft cells at comparable expression levels to the tuft marker *Dclk1* (Fig 6B). In contrast, in the mouse colon scRNA-seq *Hoxaas3* is mainly detected in goblet cells (Fig 6C). In order to examine expression in intact tissue, we performed smFISH for *Hoxaas3* in the jejunum of the mouse small intestine, which contains a relatively high fraction of goblet cells, and compared it to smFISH of the goblet cell marker *Gob5*, the tuft cell marker *Dclk1*, and *Atoh1* marking intestinal secretory precursor cells, including immature goblet and tuft cells. Based on the marker expression and the positions of the cells, we conclude that *Hoxaas3* is expressed in the early immature goblets and in the secretory precursor cells (Fig 6D). *Hoxaas3* and *Hoxa5* were occasionally co-localized, similar to the observations in HT-29 cells (Fig 6D).

scRNA-seq and smFISH from both human and mouse samples thus supports the notion that *HOXA-AS3* is expressed in a specific subpopulation, which may explain the apparently variable expression pattern that we observed in HT-29.

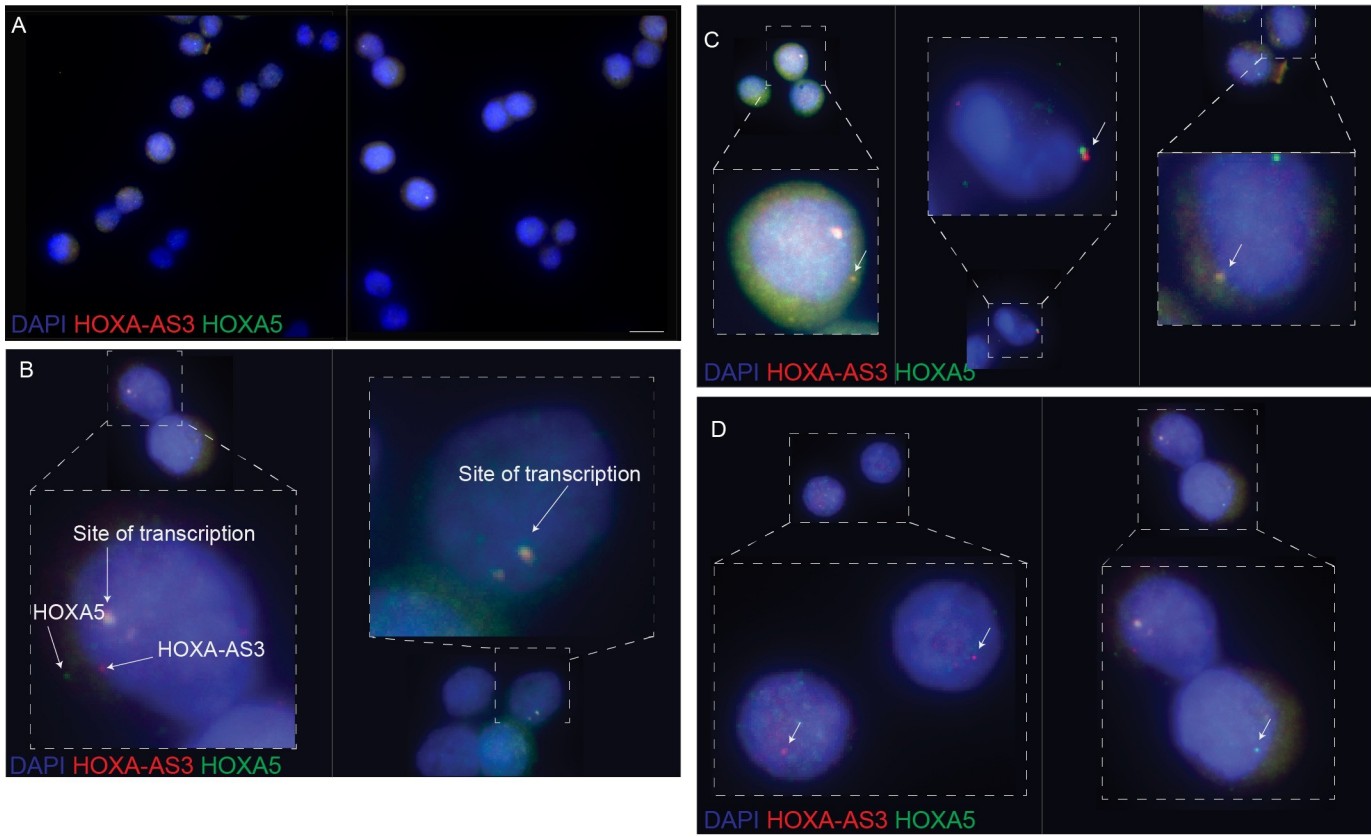

**Fig 5. Single-molecule FISH detection of *HOXA-AS3* and *HOXA5* in HT-29 cells.** (**A**) *HOXA-AS3* (red) and *HOXA5* (green) transcripts in a sample of HT-29 cells. Scale bar: 10 μm. (**B**) *HOXA-AS3* and *HOXA5* are co-localized at their presumed site of transcription. (**C**) *HOXA-AS3* and *HOXA5* are occasionally co-localized in the perinuclear area (white arrow). (**D**) *HOXA-AS3* and *HOXA5* are occasionally expressed separately.

## *HOXA-AS3* and *HOXB-AS3* are induced during early differentiation of human embryonic stem cells towards endoderm

As both *HOXA-AS3* and *HOXB-AS3* were more highly expressed in embryonic stages compared to adult tissues, we next wanted to evaluate the expression and activities of *HOXA-AS3* and *HOXB-AS3* during early developmental transitions. Endoderm is one of the three primary germ cell layers, and endoderm patterning is controlled by a series of reciprocal interactions with nearby mesoderm tissues. As development proceeds, broad gene expression patterns within the foregut, midgut, and hindgut become progressively refined into precise domains from which specific organs will arise. Human embryonic stem cells can be differentiated towards endodermal cell lineages in a robust manner, resulting, within seven days, in three different populations–anterior foregut (AFG), posterior foregut (PFG) and midgut/hindgut (MHG), using a protocol established by Loh et al. [38] (**S8A–S8C Fig**). During this differentiation process a graded, spatially collinear Hox gene expression is observed, after *in-vitro* patterning, whereby PFG cells express 3' anterior Hox genes (e.g. *HOXA1*) and MHG cells express 5' posterior Hox genes (including *HOXA10*) [38] (**S8A Fig**).

Pluripotent hESCs and cells from each stage of the differentiation were validated by multiple markers (**S4 Table**) using qRT-PCR (**S8D Fig**) and by immunostaining (**S8E Fig**), matching the expression patterns observed in the RNA-seq data from [38] (**Fig 7A**), *HOXA-AS3* and

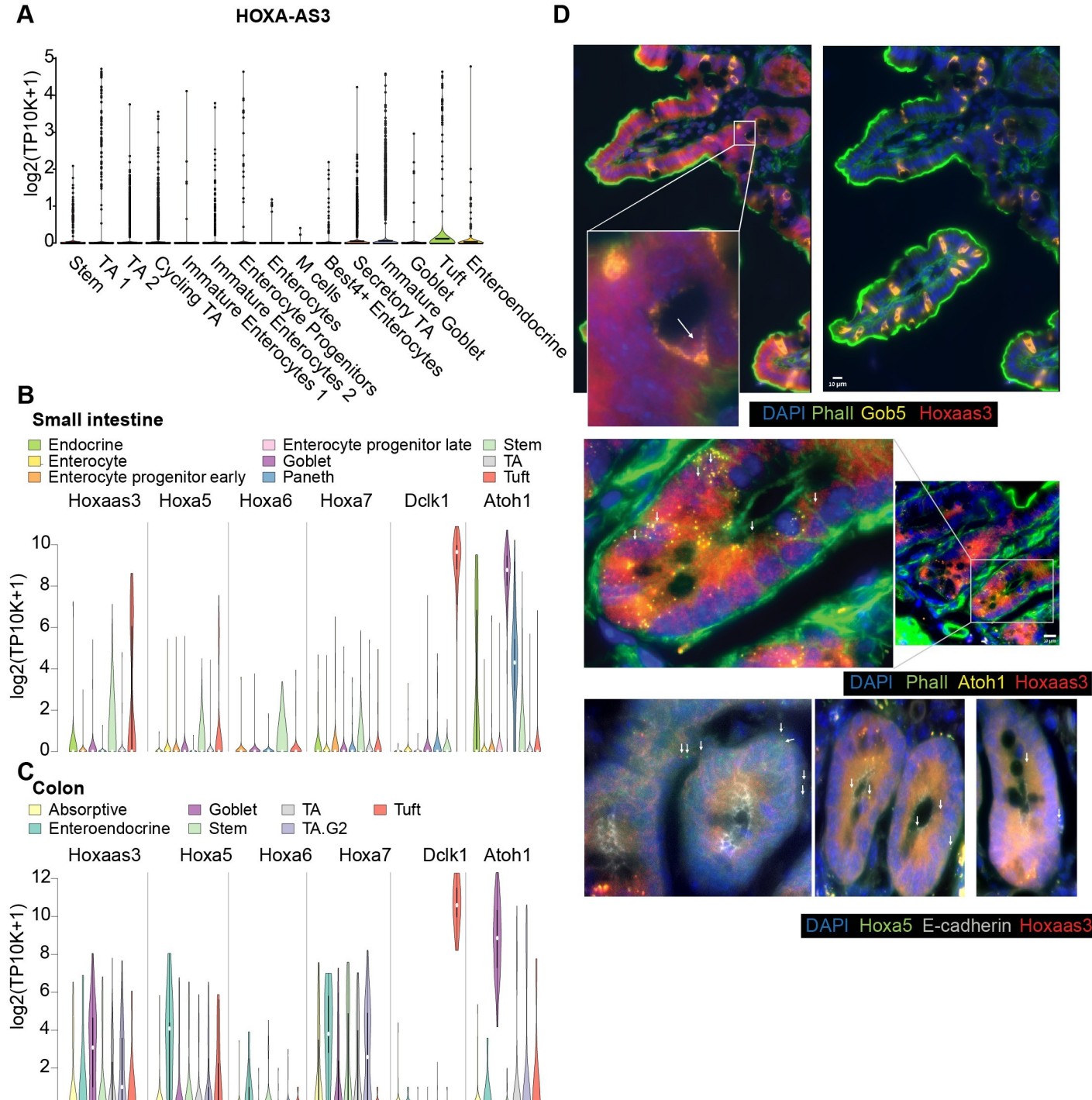

**Fig 6. Expression of HOXA-AS3 in the human and mouse gut.** (A) Expression of HOXA-AS3 in single cells of the human colon (data from [62]). **(B-C)** Expression of the indicated genes in scRNA-seq from the mouse small intestine (**B**) and colon (**C**). Data from [37]. **(D)** smFISH of *Hoxaas3*, *Hoxa5*, *Gob5* and *Atoh1* expression in the mouse intestine. Scale bar:10μm. Arrows indicate a subset of RNA molecules detected in the images.

*HOXB-AS3* were strongly induced and expressed only in the MHG population, alongside their adjacent HOX-6 and HOX-7 genes, whereas *HOXA5* and *HOXB5* were also expressed in PFG cells (**Fig 7A**).

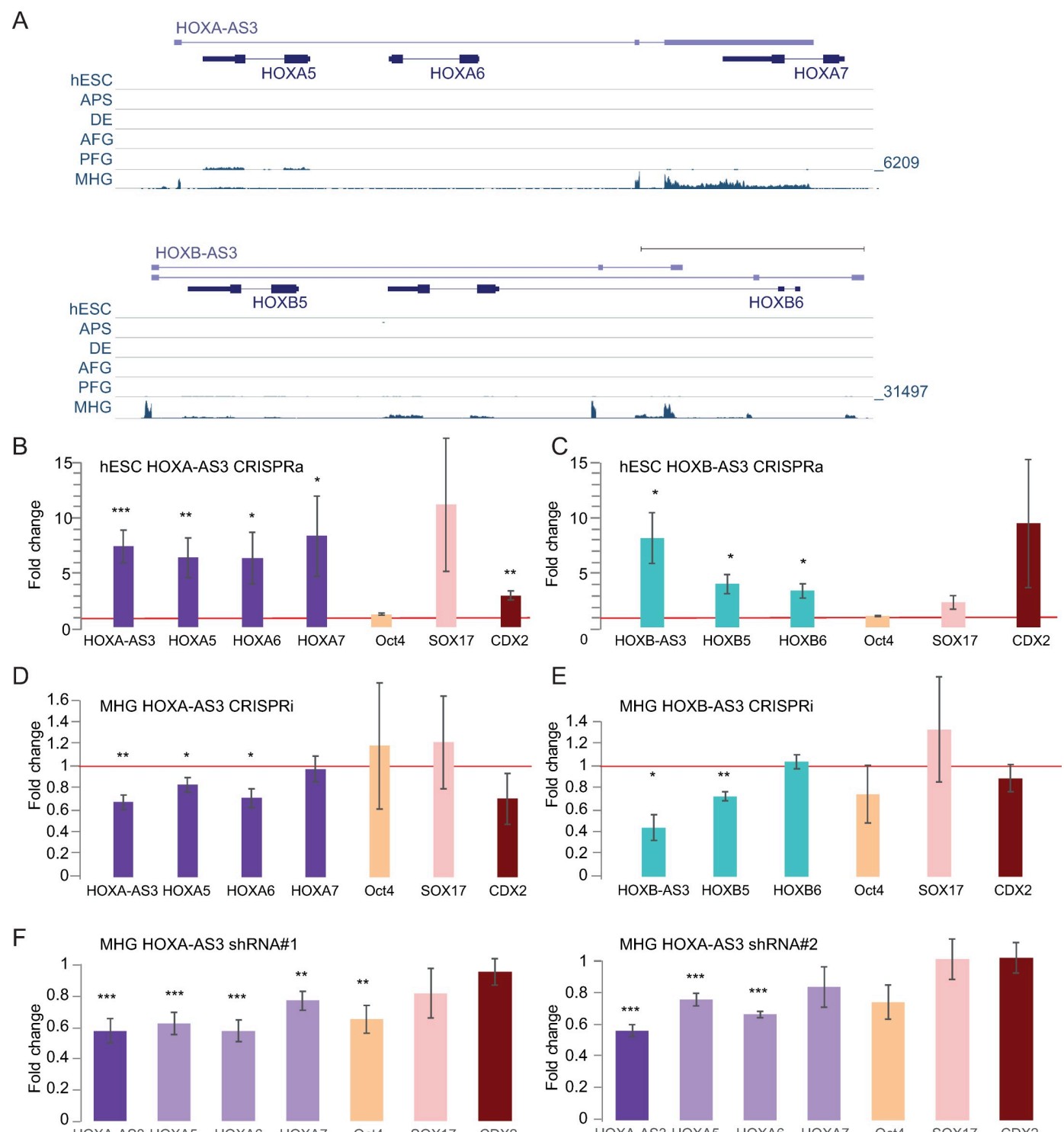

**Fig 7. Function of HOXA-AS3 and HOXB-AS3 during endodermal differentiation of hESCs. (A)** Read coverage in RNA-seq data from [38] for shown parts of the HOXA (top) and HOXB (bottom) clusters. In each cluster all the tracks are normalized together. **(B-C)** Expression levels estimated by qRT-PCR for the indicated genes in hESCs following CRISPRa-mediated 48h activation of *HOXA-AS3 (n = 7/4)* **(B)** and *HOXB-AS3* (n = 3) **(C)**. **(D-E)**. Expression levels estimated by qRT-PCR in MHG cells following CRISPRi-mediated repression of *HOXA-AS3 (n = 3)* **(D)** and *HOXB-AS3* (n = 3) **(E)**. **(F)** Changes in expression of the indicated genes following infection of hESCs with two separate HOXA-AS3 shRNAs, followed by differentiation to MHG. n = 6. *—P<0.05; **—P<0.005; ***—P<0.005. Two sided t-test. Errors bars—SEM.

### *HOXA-AS3* and *HOXB-AS3* regulate expression of their adjacent Hox genes during hESC differentiation

To study the functions of *HOXA-AS3* and *HOXB-AS3* during early steps of stem cell differentiation, we established dCas9-expressing H9 hESCs, using viral infection of Tet-dependent inducible versions of the dCAS9 and dCas9-VP64. We preferred to avoid the use of dCas9-KRAB in this system, as we were able to obtain efficient KD using dCas9 alone, which does not by itself directly affect chromatin modifications. We then established derivatives of these stable lines expressing specific gRNAs targeting the promoters of *HOXA-AS3* or *HOXB-AS3*.

After 48 hr of doxycycline (Dox) addition to the dCas9-VP64 expressing lines, we observed an up-regulation of *HOXA-AS3* and *HOXB-AS3* lncRNAs in their respective lines (**Fig 7B and 7C**). Furthermore, we observed up-regulation of the genes adjacent to these lncRNAs, even though neither of these genes are normally expressed in hESCs, and the chromatin of the HOXA cluster in hESC is in an inactive conformation [39–42]. Activation of *HOXA-AS3* in hESCs resulted in increased expression of *HOXA5-7* (**Fig 7B**). Similarly, *HOXB-AS3* activation led to an activation of *HOXB5* and *HOXB6* (**Fig 7C**). Next, we tested for changes in the pluripotency and differentiation markers in the CRISPRa lines. Although there was no remarkable change in the Oct4 pluripotency marker, we observed an increase in endodermal markers–*HOXA-AS3* or HOXB-AS3 overexpression (OE) lines led to an upregulation of *Sox17*, a definitive endoderm marker known to be required for normal development of the definitive gut endoderm [43] and in *Cdx2* levels, a marker of later stages of endodermal differentiation, expressed mainly in the MHG cells (**Fig 7C**).

Next we wanted to examine the effect of reducing the levels of *HOXA-AS3* and *HOXB-AS3* during endodermal differentiation at time points at which they are endogenously induced during the third stage of differentiation, as the cells are transitioning from DE to MHG (**Fig 7A**). For both *HOXA-AS3* and *HOXB-AS3* we obtained a ~50% KD using the Dox-inducible dCAS9, and targeting *HOXA-AS3* resulted in downregulation of *HOXA5/6/7* (**Fig 7D**), with a relatively smaller effect on HOXA7, which is generally expressed at low levels in MHG cells. KD of *HOXB-AS3* led to downregulation of *HOXB5* (**Fig 7E**). In both cases we observed no major changes in expression of markers for pluripotency (*Oct4*), endoderm (*Sox17*) and mid/hindgut (*Cdx2*) (**Fig 7D and 7E**).

In order to study the role of *HOXA-AS3* RNA product during hESC differentiation, we used the two shRNA constructs described above to generate hESC lines where *HOXA-AS3* is stably targeted by RNAi. In this system, a stable reduction of *HOXA-AS3* expression also has a similar effect on HOXA5–7 (**Fig 7E**). There was also a significant reduction in levels of Oct4, although its expression is low at the MHG stages, and so the physiological significance of this reduction is unclear. Knockdown of HOXA-AS3 and HOXB-AS3 also led to a reduction in the expression of genes from the other cluster, similar to the observations in HT-29 cells (**S8F Fig**).

## Discussion

We found here that *HOXA-AS3* and *HOXB-AS3* are ultraconserved lncRNAs which demonstrate high conservation in promoter sequence, genomic configuration and regulation that underpin similar expression patterns in specific biological processes, and relate to related functions in regulating expression of their proximal genes. The expression of *HOXA-AS3* and *HOXB-AS3* during embryonic development is particularly high in intestine-specifying lineages such as the MHG cell population that emerges during endodermal differentiation of hESCs, in the primitive streak around E7.5, in hindstomach and small intestine epithelial cells in E12 during mouse development (**S9 Fig**). Moreover we observe co-expression of *HOXA-AS3* and

*HOXB-AS3* in the adult intestine and colon in human and in mouse, specifically in cells that transition from the stem cell niche to fully specified intestinal cells in the crypt, presumably utilizing some of the same regulatory programs that are used during early development. smFISH in mouse intestine showed specific enrichment of HOXA-AS3 expression in early immature goblet cells and in the secretory precursor cells, highlighting the expression timing to mid-differentiation–the phase where the cells are committing and acquiring their specific fate, in concurrence to its induction in hESC differentiation, and potentially related to expression in only a small subset of HT-29 cells in culture.

There have been several recent reports about the functions of *HOXA-AS3* and *HOXB-AS3* in other systems. *HOXA-AS3* was reported to be induced during adipogenic induction of human mesenchymal stem cells (MSCs), and its silencing promoted proliferation of MSCs and inhibited osteogenesis *in vitro* and *in vivo*, in both human and mouse cells [44]. Positive effects of *HOXA-AS3* on proliferation and migration *in vitro* and during tumorigenesis *in vivo* were also observed in glioma cells [33]. Another study found that *HOXA-AS3* promoted proliferation, migration and invasion in A549 lung carcinoma cell line and tumor growth *in vivo* [27], where it was found in both the nucleus and the cytoplasm, consistent with our data in HT-29 cells. In that study *HOXA-AS3* was suggested to positively regulate *HOXA6*, as siRNA-mediated KD of *HOXA-AS3* reduced levels of *HOXA6* mRNA and protein (but not those of *HOXA5*) in A549 cells. A more recent publication extended the positive effects of *HOXA-AS3* on proliferation to additional non-small-cell lung carcinoma cell lines [34]. These studies are overall consistent with our observations that in normal tissues *HOXA-AS3* is preferentially expressed in proliferating progenitors, and the reduction in a proliferation signature in the RNA-seq data of *HOXA-AS3* KD cells. Lin et al. found *HOXA-AS3* to have a negative effect on *HOXA3* expression, by binding to both *HOXA3* mRNA and *HOXA3* protein. Lastly, *HOXA-AS3* was recently proposed to regulate NF-kappaB signalling in HUVECs [26]. Notably, in ENCODE RNA-seq data *HOXA-AS3* is undetectable in both A549 cells and HUVECs (which do express *HOXA6* and *HOXA7*), whereas it is well-expressed in the HT-29 cells we used in this study (**S5 Fig**).

*HOXB-AS3* was reported to be down-regulated in colorectal cancers and to produce a 53 aa protein conserved in primates [17]. Notably, PhyloCSF scores throughout *HOXB-AS3* are negative (**S1 Fig**), so it is very unlikely that it encodes a conserved protein. In colorectal cancer cells *HOXB-AS3* was shown to inhibit cell proliferation [17]. In NPM1-mutated acute myeloid leukemia cells, *HOXB-AS3* does not associate with polysomes and promotes cell proliferation in both human and mouse leukemia cells [35,45]. Interestingly, in this system, KD of *HOXB-AS3* using antisense oligonucleotides did not affect expression of other Hox genes, but rather regulated expression of ribosomal RNA, in *trans*, via interaction with EBP1 [35], consistently with our observation of changes in rRNA processing genes upon HOXB-AS3 KD in HT-29 cells. There is therefore evidence of trans-acting activities of *HOXB-AS3*. We note that our findings about cis-acting regulation of *HOXB6* and *HOXB7* by *HOXB-AS3* do not exclude these additional functions and in fact it is likely that lncRNAs that are robustly expressed and highly conserved have aquired additional, species- or clade-specific functions during evolution.

We report a positive effect of *HOXA-AS3* and *HOXB-AS3* production on the expression levels of their overlapping HOX5–7 genes. We studied these effects in vitro in cultured cells and mostly in a cancer cell line with an abnormal karyotype, and future studies will elucidate the roles of HOXA-AS3 and HOXB-AS3 RNA products *in vivo*. Notably, the positive effect we report does not translate into a tight co-expression between the lncRNAs and the protein-coding genes in this region when considering a broad range of conditions and cell types (**Figs 2A** and **S4A**), likely because other mechanisms contribute to expression of the HOX5–7 genes in

cells which do not express the lncRNAs. For example, we see strong expression of HOXA5 in the PFG cell population in differentiating hESCs (**Fig 7A**). Various mechanism *for* cis-acting regulation of gene expression by lncRNAs have been demonstrated in different systems [46]. Future studies will elucidate the mechanism underlying the regulation of HOX5–7 gene expression by *HOXA-AS3* and *HOXB-AS3*, which may resemble those of other lncRNAs. It is of particular interest to study whether HOXA-AS3 and HOXB-AS3 influence the nature of the transcripts produced in the complex loci of the HOX clusters, e.g., but influencing promoter choice. Genome editing of the loci can be particularly powerful for promoting understanding of lncRNA biology, but it is particularly difficult to perform and interpret in the Hox clusters, due to the high density of gene regulatory elements within the clusters and the complex relations between them. The most relevant systems to perform editing of the human *HOXA-AS3* and *HOXB-AS3* is likely hESCs, which can then be differentiated to MHG cells, but CRISPR-mediated editing in hESCs is inefficient [47]. Indeed, despite screening hundreds of clones, we were so far unsuccessful in obtaining homozygous deletions of the HOXA-AS3 promoter in hESCs. Mouse models carrying specific manipulations, such as insertion of polyA sites, will also be highly informative.

Some of the protein-coding genes and miRNAs in the Hox clusters were shown to be functionally equivalent to each other and to contribute differentially to organismal function via their divergent expression patterns [48,49]. These orthologs formed by duplication during the formation of the four vertebrate Hox gene clusters. The paucity of known lncRNA paralogs present in different Hox clusters can be rather easily explained by the overall high rate of lncRNA evolution [50], which likely rewired the sequences and exon-intron architectures of Hox lncRNAs extensively over the past 500 million years. The numerous features we identified as shared between *HOXA-AS3* and *HOXB-AS3* suggest that at least some lncRNAs were duplicated and maintained regulatory functions in the Hox cluster throughout vertebrate evolution, during which individual clusters also acquired additional lncRNAs, some of which are functional, and that further sculpted gene expression within each cluster. Importantly, there is also evidence of extensive cross-regulation between the clusters, including by lncRNAs [10]. Future studies will examine the potential contribution of *HOXA-AS3* and *HOXB-AS3* lncRNAs to cross-cluster regulation, as well as the extent of similarity that they maintained in their modes of action.

## Materials and methods

### Tissue culture

H9 hESC were routinely cultured on irradiated MEFs in hESC medium: DMEM/F-12 (Sigma, D6421), 15% KNOCK-OUT Serum Replacement (Gibco, 10828–028), Glutamax X1, (Gibco, 35050–038), 1% Non-essential amino acids (NEAA) (Biological Industries, 01-340-1B), 0.1 mM 2-mercaptoethanol (Gibco, 31350–010), and 8ng/ml bFGF (Peprotech, 100-18B), at 37°C in a humidified incubator with 5% CO2. HT-29, MCF7 and HEK293T cell lines and were routinely cultured in DMEM containing 10% fetal bovine serum and 100 U penicillin/0.1 mg ml$^{-1}$ streptomycin, at 37°C in a humidified incubator with 5% CO2.

### Endodermal differentiation

Endodermal differentiation was performed as previously described [38]. Pluripotent human stem cells were grown in the absence of MEF for four passages in mTeSR1 (StemCell Technologies, 85850) and seeded on Geltrex (invitrogen, A1413202). After 1–2 days of recovery in mTeSR1, hESC were washed with F12 (Gibco, 21765–029) and then were treated for 24 hours with Activin A (100 ng/mL, R&D Systems, 338-AC-010), CHIR99021 (2 μM, Stemgent,

04–0004), and PI-103 (50 nM, Tocris, 2930) in CDM2 to specify APS. Afterwards, cells were washed (F12), then treated for 48 hours with Activin A (100 ng/mL) and LDN-193189/ DM3189 (250 nM, Stemgent, 04–0074) in CDM2 to generate DE by day 3. Day 3 DE was patterned into AFG, PFG, or MHG by 4 days of continued differentiation in CDM2. DE was washed (F12), then differentiated as follows: AFG, A-83-01 (1 μM, Tocris, 2939) and LDN-193189 (250 nM, Stemgent, 04–0074); PFG, RA (2 μM, Sigma, R2625) and LDN-193189 (250 nM); MHG, BMP4 (10 ng/mL, R&D Systems, 314-BP-010), CHIR99021 (3 μM, Stemgent, 04–0004), and FGF2 (100 ng/mL, Peprotech, 100-18B), yielding day 7 anteroposterior domains. Media was refreshed every 24 hours for each differentiation step.

## HT-29 enterocytic differentiation

Enterocyte differentiation was performed as previously described [51]. HT-29 cells were seeded in 90% confluence on ThInCerts (Greiner, 60–657641) in 6 well plates. Cells were cultured for 31 days in glucose free conditions (Sigma, 11966–025) and the medium was changed every 2 days.

## Transfections

Plasmid transfections for HEK293T, MCF7 and HT-29 were performed using PolyEthylene Imine (PEI) (PEI linear, M*r* 25,000, Polyscience). CRISPRi/a transient experiments were harvested after 72h. siRNAs were transfected into HT-29 cells at 25 nM siRNA pool or with control pool (Dharmacon) by using DharmaFECT 4 (horizon, T-2004) following the manufacturer's protocol. Cells were harvested after 48h of siRNA treatment.

## Lentivirus production and stable lines generation

All lentivirus production was performed as previously described [52]. Medium was collected from plates 72 hr after transfection, filtered by VIVASPIN (Sartorius, VS2001), concentrated and stored –80˚C. hESC and HT-29 cells were infected by lentiviral particles incubated in the growth medium containing and 8μg/ml Polybren (Sigma, 107689) to attached cells, following selection after 24h for several passages for pool isolation.

## RNA and RT-qPCR

Total RNA was extracted from different cell lines and mouse tissues, by using RNeasy (Qiagen) according to the manufacturer's protocol. cDNA was synthesized by using qScript Flex cDNA synthesis kit (Quanta, 95049). Fast SYBR Green master mix (Life, 4385614) was used for qPCR with gene-specific primers (**S5 Table**).

## Immunofluorescence

Cultured cells were fixed with 4% paraformaldehyde for 10 minutes. Fixed cells were permeabilized using 0.1% triton X-100, blocked with 5% normal goat serum, incubated with a primary antibody, followed by incubation with a secondary antibody conjugated to a fluorescent dye. Antibodies used: Rabbit α-Eomes (Abcam, ab23345), Goat α-Sox17 (R&D Systems, AF1924), Goat α-Cdx2 (R&D Systems, AF3665), Goat α-Otx2 (R&D Systems, AF1979).

## Single-molecule FISH

Cultured cells were fixed with 4% paraformaldehyde 24 hr after plating. Tissue was frozen in Tissue-Tek O.C.T compound (Sakura 4583) blocks and sectioned using a Leica cryostat (CM3050) at 10 μm thickness. Libraries of 96 and 94 probes (**S3 Table**) were designed to target

human *HOXA-AS3* and mouse *Hoxaas3* RNA sequences, respectively and a commercially available library of 48 probes was used to detect HOXA5 (cat # VSMF-2538-5) (Stellaris RNA FISH probes, Biosearch Technologies). Hybridization conditions and imaging were as previously described [53,54]. smFISH imaging was performed on a Nikon-Ti-E inverted fluorescence microscope with a 100 × oil-immersion objective and a Photometrics Pixis 1024 CCD camera using MetaMorph software as previously described [55].

## RNA-seq

HT-29 cells were transfected with 25nM siRNA against *HOXA-AS3*, *HOXB-AS3*, or with control siRNA using DharmaFECT 4 transfection reagent. RNA was extracted using TRIREA-GENT (MRC TR 118) 48 hours post transfection, 1µg of total RNA was used for RNAseq library preparation using the SENSE mRNA-Seq Library Prep Kit V2 for Illumina (Lexogen, LX-001.96) according to the manufacturer's recommended protocol. Gene expression levels were quantified using RSEM [56] and a RefSeq gene annotation database. Differential expression was computed using DESeq2 with default settings [57]. RNA-seq datasets are deposited in GEO database under the accession GSE168444. RNA-seq data from previous studies were downloaded from the SRA database, and quantified using RSEM with the same annotation file. Gene Ontology enrichment was analyzed using GORilla [32] on gene lists sorted by DESeq2 log2FoldChange, on genes with an average FPKM larger than 1, after excluding pseudogenes, and transcripts shorter than 200 nt.

## gRNA cloning

Guide RNAs were designed by CHOPCHOP. For single sgRNA expression guide sequences were cloned into pKLV-U6gRNA(BbsI)-PGKpuro2ABFP (Addgene plasmid #50946) [58]. following Zhang Lab General Protocol (https://media.addgene.org/cms/filer_public/6d/d8/6dd83407-3b07-47db-8adb-4fada30bde8a/zhang-lab-general-cloning-protocol-target-sequencing_1.pdf). For dual sgRNA expression a mega-primer donor was generated by PCR using primers with the following structure:

Fw primer: `tacatcttgtggaaaggacgaaacaccg-gRNA1-gttttagagctagaaat agcaagttaaaataaggc`

Rev primer: `cttgctatttctagctctaaaac-gRNA2(rev-compliment)-gggaaa gagtggtctcatacagaacttataag`

with pDecko-GFP (Addgene plasmid #72619 [59]) as template.

The PCR product was cloned into pDecko-GFP by restriction free cloning [60].

## Supporting information

**S1 Fig. CAGE and RNA-seq read coverage support for gene models of HOXA-AS3 and HOXB-AS3 orthologs. (A)** Human gene models annotated in GENCODE v36 in the central part of the HOXA and HOXB clusters (protein-coding transcripts are in blue and noncoding are in green). The total CAGE read coverage from FANTOM5.5 is shown on top. **(B)** In each species, annotated or reconstructed gene models are shown for *HOXA-AS3* or *HOXB-AS3* (the strand from which they are produced is defined as the '+' strand) and the protein-coding HOX5–7 genes (transcribed from the '-' strand). RNA-seq data are from the following datasets: SRP023152 (opossum), SRP041863 (Chicken), GSE136018 (Medaka), and SRP013772 (Shark). (EPS)

**S2 Fig. PhyloCSF scores for the HOXA-AS3 and HOXB-AS3.** PhyloSCF scores [16] taken from the PhyloCSF UCSC genome browser, for each of the three frames for the '+' strand

from which HOXA-AS3 and HOXB-AS3 are transcribed. Position of the proposed ORF from [17] is shown.
(EPS)

**S3 Fig.** **(A)** Sequence conservation and similarity in the *HOXA-AS3* and *HOXB-AS3* promoter regions. Exonic sequences (where known) are in bold. Predicted binding sites of the indicated transcription factors, taken from the UCSC genome browser are shaded in yellow. Regions of the 5' splice sites at the end of the first exon, where known, are shaded in blue. **(B)** Number of CDX1 or CDX2 binding sites predicted by JASPAR [63] in 201,802 human promoters annotated in FANTOM5.5 [19]. For each TSS we considered the region -100 to 100 relative to the TSS. Selected genes are highlighted.
(EPS)

**S4 Fig.** **(A)** Distribution of correlation coefficients between expression patterns of pairs of genes within the same Hox cluster (red) and found in different Hox clusters (blue). The correlation coefficient between *HOXA-AS3* and *HOXB-AS3* is shown in green. All the coefficients are computed across all the samples from the human FANTOM5.5 data. **(B)** as in Fig 2A for the mouse FANTOM5.5 data. **(C)** Expression of *Hoxaas3* in clusters of single cells from the Tabula Muris database [64]. Eight cell groups with the highest expression are shown.
(EPS)

**S5 Fig. Spatial expression patterns of *HOXA-AS3* and *HOXB-AS3* and other genes in the E7.5 mouse embryo.** Geo-seq data were re-mapped to the RefSeq annotations and visualized as in [20]. Each gene is shown on a separate scale. Genes are grouped based on their genomic location or gene family.
(EPS)

**S6 Fig. Expression of central HOXA and HOXB genes in ENCODE cell lines.** **(A)** Shown are selected RefSeq gene models for the indicated genes alongside RNA-seq strand-specific read coverage from the indicated cell lines from ENCODE datasets of total (HT-29) or polyA-selected (A459, HUVEC) RNA on the indicated strand as depicted in the UCSC genome browser. *HOXA-AS3* and *HOXB-AS3* are transcribed from the '+' strand and the protein-coding genes from the '-' strand. **(B)** Read coverage from the ENCODE datasets in IGV genome browser, showing the agglomerated coverage from both strands, and the splice-junction-supporting reads from the '+' strand (blue) and the '-' strand (red).
(EPS)

**S7 Fig. Cross-regulation by HOXA-AS3 and HOXB-AS3 of HOXB and HOXA clusters.** **(A)** Expression levels in HT-29 cells of the protein-coding genes in the indicated paralogous HOX gene group. Shown are the average expression levels across our RNA-seq dataset. **(B)** As in Fig 4A and 4B, for the indicated genes from the HOXB (left) and HOXA (right) clusters. **(C)** As in Fig 3A and 3B, for the indicated genes in the HOXB (left) and HOXA (right) clusters. **(D)** Changes in gene expression and DESeq2 p-values for the transcriptome-wide changes in gene expression following siRNA-mediated KD of HOXA-AS3 (left) and HOXB-AS3 (right) in HT-29 cells. Genes with adjusted P<0.05 are in red.
(EPS)

**S8 Fig. hESC endodermal differentiation overview.** **(A)** Expression levels of Hox genes and lncRNAs in data from [38]. **(B)** Endodermal differentiation process and signaling molecules. **(C)** Characterization of different cell morphology in different stages of the differentiation. **(D)** RNA levels and expression dynamics were measured by qRT-PCR at different stages of endodermal differentiation, and normalized to actin. **(E)** Immunofluorescence stainings of human

ESCs differentiated in different stages of differentiation. **(F)** As in Fig 7E (left) and Fig 7F (right) for the indicated genes.
(EPS)

**S9 Fig. Regulation of *HOXA-AS3* and *HOXB-AS3* by CDX1/2 transcription factors.** CDX2 ChIP-seq read coverage from the gut at the indicated stage and RNA-seq at E12 in the indicated tissue, data from [65]. The region corresponding to the promoters of *Hoxaas3* and *Hoxb5os* is shaded.
(EPS)

**S1 Table. FANTOM5 expression data.**
(XLSX)

**S2 Table. RNA-seq data and GO enrichments.**
(XLSX)

**S3 Table. Markers used for validation.**
(XLSX)

**S4 Table. smFISH probes.**
(XLSX)

**S5 Table. Primers and siRNAs.**
(XLSX)

**S1 Dataset. LncLOOM analysis of the HOXB-AS3 conservation.**
(ZIP)

## Acknowledgments

We thank members of the Ulitsky lab for useful discussions, Thomas Toubul, Peter DeHoff, and Louise Laurent for discussions on the use of CRISPRi and CRISPRa in hESCs, Gilad Beck for stem cell advice and valued contributions to the hESC work, and Shani Ben-Moshe for help with smFISH of intestinal markers and mouse intestine sections.

## Author Contributions

**Conceptualization:** Neta Degani, Igor Ulitsky.

**Data curation:** Igor Ulitsky.

**Formal analysis:** Neta Degani, Igor Ulitsky.

**Funding acquisition:** Igor Ulitsky.

**Investigation:** Neta Degani, Yoav Lubelsky, Rotem Ben-Tov Perry.

**Methodology:** Neta Degani, Yoav Lubelsky, Rotem Ben-Tov Perry, Elena Ainbinder.

**Project administration:** Igor Ulitsky.

**Supervision:** Igor Ulitsky.

**Visualization:** Igor Ulitsky.

**Writing – original draft:** Neta Degani, Igor Ulitsky.

**Writing – review & editing:** Neta Degani, Igor Ulitsky.

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
