## [Decision Letter · Decision Letter 0]

20 Jan 2021

Dear Dr Ulitsky,

Thank you very much for submitting your Research Article entitled 'Highly conserved and cis-acting lncRNAs produced from paralogous regions in the center of HOXA and HOXB clusters in the endoderm lineage' to PLOS Genetics.

The manuscript was fully evaluated at the editorial level and by independent peer reviewers. The reviewers appreciated the attention to an important topic but identified some concerns that we ask you address in a revised manuscript.

We therefore ask you to modify the manuscript according to the review recommendations. Your revisions should address the specific points made by each reviewer.

[LINK]

Yours sincerely,

Eric A Miska, PhD

Associate Editor

PLOS Genetics

Bret Payseur

Section Editor: Evolution

PLOS Genetics

Reviewer's Responses to Questions

**Comments to the Authors:**

Reviewer #1: The manuscript by Degani et al presents the analysis of the expression and regulation of two antisense lncRNAs within the Hox cluster. The study presented is of interest as the much remains to be learned regarding the function and impact of lncRNAs. The interest of the study stems from the high conservation of these two lncRNAs, their impact on Hox gene expression, but most importantly by the approach developed by the authors combining mining existing omic data and carefully designed experiments assessing the regulation of Hox genes by these lncRNAs and allowing to properly test transcription vs transcript mediated effect. The work is presented in a succinct, clear manner and is discussed in context of the current state of knowledge.

I have only minor comments

In the discussion it would be good to contrast the results shown on figure 2a and 2b with no or negative correlation between HOXA-AS3, HOXB-AS3 and the other HOX genes and the results stemming from the analyses of HT-29 cell as they appear somewhat conflicting. This is most likely the consequence of the tissues available and cell population represented in the data used for Figure 2.

- FigS1 - The labelling of strand and directionality for each of the species needs to be clarified. Perhaps more information in the legend, and a mention of the species-specificity of the strandedness where it is mentioned in the introduction. It is currently a little confusing as the figures seem to contradict the text without clarification that strandedness varies in the species presented.

- Fig S1: can the authors check the tracks for the RNA-Seq and the annotations for the opossum, they are not aligned - spelling on the shark track - Kindey

- Fig2 - the expression is of these lncRNAs is very low and the TPM>=1 threshold is understandable, but how robust are these findings to raising that threshold?

- Fig S5 shows very large transcriptional activity on both strands, I would be careful in drawing conclusions here. To reduce the noise could you plot the splicing junctions? as the exons are clearly marked especially for HOXB-AS3?

Page 9: “These results suggest that HOXA-AS3 and HOXB-AS3 production or their RNA products have a positive regulatory effect on the expression of the neighboring genes HOX5–7 genes.” The authors do not show significant effect of activation for HOXB-AS3 it would be good to slightly change the phrasing to also include this observation.

- Fig 6D - the arrows for HOXA-AS3, are these exhaustive? It looks like there are other molecules (including on the edge of next cell in the green "Phall region" top left of image). Do the different arrows colours represent anything? They are not referred to in the manuscript anywhere? Refer to them in the manuscript or remove them.

Reviewer #2: The authors of the study "Highly conserved and cis-acting lncRNAs produced from paralogous regions in the center of HOXA and HOXB clusters in the endoderm lineage" focus on a lncRNA derived antisense to HOXA3 in several species. Using their expertise in evolutionary conservation they start by finding syntenic regions across multiple species comprised of an array of evolutionary distances. The PhyloCSF analysis indicates that in most species this lncRNA is not likely to encode a protein (where as the positive control of HOX exons show clear synonymous mutations throughout evolution expected for proteins). Deeper sequence analysis reveals a tandem CDX1/2 sites that are very conserved. The authors perform co-expression analysis and find these two loci are highly correlated in expression. Moreover, it is expected that in IMR90 that only express proximal HOX genes that these genes would be expressed -- the authors find the same for HT-29 cancer line (where as a primary fibroblast line may have been more beneficial for normal HOX biology and a system developed by Howard Chang's lab (Rinn et al. PLoS Genetics 2006, Figure 7). The authors used the HT-29 line to perform LOF studies using CRISPR-I. They observe that the neighboring genes in both cases are down-regulated upon LOF of the lncRNA. Similarly, GOF results in increased expression of neighboring genes. The authors continue to show that HOXA3-as is expressed in specific regions of the intestine, but not relative to the HOXB-as. Moreover, the authors investigate the roles of these lncRNAs in hESC differentiation and find they are specifically expressed in mid/hind gut (MHG). The authors then perform CRISPR based LOF/GOF and see significant but very small effect sizes in regulation of MHG differentiation. Finally, the authors explore the transcriptional regulation of these Antisense lncRNAs by CDX1/2. They show that depletion of CDX1 down-regulates both HOXA3/B-as lncRNAs.

Overall, the evolutionary analysis of this study is interesting and the co-expression compelling. Yet all the LOF/GOF studies are far less convincing, mostly owing to highly variable and or small effect sizes. I have the following concerns before high-profile publication in PLoS Genetics.

1) How often does the CCATAA motif show up in randomly selected transcripts? Can the authors determine if this is unique to the HOXA/B-as genes relative to all other HOX genes?

2) HOXA/B-as lncRNAs are highly correlated. If all hox pairs are considered is the correlation more than expected for other hox genes that are highly correlated?

For example HoxA1-7 should be highly correlated with HOXD1-7. Essentially one would expect a high correlation between the binary proximal and distal information expressed in each hox cluster. This was found by the Duboule lab in vivo as well. Counter to the co-linearity model it is noted in this and other studies that the hox cluster either expresses HOX1-7 or HOX9-13 paralogs (at least for HOXA and D clusters). Thus, it would not be surprising to find strong correlation across HOX clusters since there are typically only binary patters on prox-on or distal-on but not prox & distal on (as would have been suggested by the collinearity model).

Similarly, Is HOTAIRM1 as or more correlated with HOXA1 & HOXA3 as HOXB3-as and HOXA3-as pairs? Or how does the HOXA3/B3-as correlation compare to intra-cluster correlation? This could be done systematically with all HOX pairs and determine the observed value of HOXA3/B3 relative to all other correlations -- is the observed significant relative the expected empirical null?

3) The authors show that LOF and GOF decrease and increase the neighboring genes respectively. I may have missed it, but did the authors check if there was cross talk between the HOXA/B LOF/GOF experiments? For example does LOF/GOF of HOXA3-as affect HOXB? Vice-versa? From my reading, when HOXA3-as CRISPR-I/A was performed the authors only looked at the neighboring genes in cis -- where as it would seem important to see if HOXA3-as also affected HOXB-as. Same for Figure 7.

4)Minor: It is somewhat concerning that HT-29 cell lines have low expression and extra copies of HOX clusters -- where as IMIR90 would not have this issue.

5) RNA-FISH wouldn't one expect to see more cytoplasmic localization of HOXA5 as a positive control for spatial resolution of RNA-FISH?

6) Intestine FISH would be good to see HOXA3-as and HOXB-as co-localized with target genes as in Figure 5?

7) Figure 7: everything in this figure trends the right way, but differences that are significant are very small in effect-size. While the HOXA3-as LOF/GOF seems ok in HESCs the depletion in MHG shows ~40% depletion of HOXA3-as and 20-30%% depletion of HOXA5. Whilst the non-significant OCT4 and SOX17 markers have a variance that encompasses the effect size of the "significant" changes. Does this have physiologically visible phenotype? For example, can the authors quantify using FISH approaches in previous figures that these markers and or genes are making a significant change in differentiation? Again is there cross-talk as the qRT-PCR only focuses on neighboring genes of LOF/GOF target and not the reciprocal cluster?

8) Surprising that the combo of CDX 1/2 has less of an affect than either alone?

Reviewer #3: Hox cluster regulation has long been a central setting with which to reveal and dissect mechanisms of gene/genome regulation. Assessment of lncRNAs within Hox clusters have not been without significant controversy (Hotair), and the continued identification and functional assessment of Hox-embedded ncRNas is important to understand their quantitative impact in shaping Hox output and in revealing commonalities/differences in the mechanisms by which they control Hox cluster expression.

In this manuscript by Degani and colleagues demonstrate i) genomic conservation of two Hox-embedded lncRNAs across vertebrate species; ii) utilise published resources to good effect to characterise the developmental and adult expression of these lncRNAs with additional fluorescent in situ hybridisation to provide spatial detail; iii) perform in vitro functional analyses indicating these lncRNAs have a positive effect on Hox gene expression of the opposite strand. Overall, the manuscript is quite straightforward, well written, I have no major technical concerns with minor technical comments below. Greater understanding of global Hox impact, and assessment of any altered ESC differentiation endpoint, dowstream of altered Hox signatures, would strengthen the impact of this work.

Genomic and transcriptomic analyses:

The transcript assessment in Figure 1 looks simplified. For example, we know the Hoxa5/a6 locus is highly complex, with various transcripts produced (see work of Lucie Jeanotte). I understand this figure does not need to reflect that level of complexity in Hox protein coding genes, but is there any evidence of alternate antisense transcripts produced? What do the opening words “in human” mean – ie, what cumulative datasets is this data derived from? ( I later see in Supp fig 2 there are 2 variants? … and Supp fig 5 that Hoxb-AS3 does appear 3 transcripts in cell lines). Please clarify.

I believe the authors mean orthologue rather than homolog throughout when comparing species?

Pg 4:

“The corresponding positions of the two lncRNAs and their high conservation in other species made us scrutinize…” Just to be clear, please add… the high conservation of their presence in other species (as written it could lead reader to think this was sequence conservation which I don’t believe was assessed directly). Regarding this last point, is there any evidence of sequence conservation?

The single cell analysis is a good addition. The relatively higher expression of Hoxaas3 in NMP, caudal epiblast, caudal mesoderm would suggest these antisense transcripts are expressed similar to Hox protein coding genes in the overall whole embryo A-P context. Whole mount ISH of each antisense transcript would be a good addition here, particularly as this data indicates the antisense transcripts are more highly expressed than the protein-coding genes they overlap.

Functional and expression analyses:

Elegant strategies employed, in some cases multiple strategies to corroborate.

For each gain or loss-of-function in vitro perturbation, a clear effect was observed for the Hox genes overlapping the antisense transcript in question (a5/a6/a7 for Hoxasa3). This manuscript would greatly benefit from a more comprehensive assessment of Hox genes both cis and trans, particularly given later the authors show it is the RNA transcript itself that is functional, not simply a local chromatin mechanism.

Figure S6: MCF7 cell line, lncRNA activation – this experiment was done twice, with no indication of whether the Hox protein coding activation is significant (for a6 and a7 at least). Please repeat with stats or remove from manuscript.

Regarding Figure 6, I find it quite difficult to interpret image 6D. There appears to be a haze of red signal indicating Hoxaas3, however it is indicated in the text and from scRNAseq data to be restricted to certain cell types – this is not apparent. Can the authors please clarify, and describe what technical controls are performed.

The ESC differentiation section was an excellent addition, strongly suggest to comprehensively characterise Hox expression in this system.

The Cdx1/2 direct regulation of HoxAS transcripts is interesting but of course preliminary. I find it difficult to interpret FigS8 (B) – siCdx1 results in expected knockdown of Cdx1, but siCdx2 also does, and when both siRNAs used the level of knockdown is less than either alone? Similar questions with other graphs. The benefit of its inclusion in general, and moreover, without directly supportive ChIPseq data is questionable.

Very minor text points

Abstract

“Sequence-similar homologs of both lncRNAs are found in multiple vertebrate species.” As mentioned above, I believe this is meant to be orthologs, and second, I believe you have compared the promoter sequence but have you compared the lncRNA sequence similarity to support this statement?

Pg 2:

“the molecular pathways that dictate their collinear expression remain mostly unknown.” Not sure this is strictly true, there’s increasing work in both ESCs and in vivo showing the signals and mechanisms that guide correct temporal Hox activation. This is of course not the point of this ms, but I would just slightly reword.

Pg2:

“miR-196 (iab-4 in D. melanogaster)”

these microRNAs show functional conservation, but they are actually not conserved in sequence, nor located at the exact syntenic position, so are not related.

**Have all data underlying the figures and results presented in the manuscript been provided?**

Reviewer #1: Yes

Reviewer #2: Yes

Reviewer #3: Yes

PLOS authors have the option to publish the peer review history of their article (what does this mean?). If published, this will include your full peer review and any attached files.

Reviewer #1: No

Reviewer #2: **Yes: **John Rinn

Reviewer #3: **Yes: **Edwina McGlinn

---

## [Decision Letter · Decision Letter 1]

23 Apr 2021

Dear Dr Ulitsky,

Thank you very much for submitting your Research Article entitled 'Highly conserved and cis-acting lncRNAs produced from paralogous regions in the center of HOXA and HOXB clusters in the endoderm lineage' to PLOS Genetics.

The manuscript was fully evaluated at the editorial level and by independent peer reviewers. The reviewers appreciated the attention to an important topic but identified some concerns that we ask you address in a revised manuscript.

We therefore ask you to modify the manuscript according to the review recommendations. Your revisions should address the specific points made by each reviewer.

[LINK]

Yours sincerely,

Eric A Miska, PhD

Associate Editor

PLOS Genetics

Bret Payseur

Section Editor: Evolution

PLOS Genetics

Reviewer's Responses to Questions

**Comments to the Authors:**

Reviewer #1: The authors have addressed appropriately the comments raised previously

Reviewer #3: The revised manuscript is very good, the varioius transcriptomic analyses and figures incorporated greatly strengthen the ms and taking out the more preliminary work focuses the story. I still would have loved to see the detailed characterisation of ESC differentiation but comprehensive analysis in another cell line is sufficient. I have no further concerns.

Reviewer #4: The manuscript by Degani and co-authors presents an evolutionary and functional analysis of two long non-coding RNAs, transcribed on the antisense strand in the HOXA and HOXB clusters. The authors show that these lncRNAs are highly conserved during evolution and that their origin likely predates the whole-genome duplications that led to the emergence of the HOX gene clusters. They perform computational analyses of publicly available gene expression resources (including both bulk tissue expression and single cell RNA-seq data), as well as their own expression experiments to determine the spatial and temporal expression patterns of these two lncRNAs and of the neighboring HOXA and HOXB genes. Moreover, they perform in vitro knockdown (with CRISPRi and siRNA) and activation experiments for these two lncRNAs and they observe an up-regulation of the neighboring HOXA5-7 and HOXB5-7 genes. They conclude that these lncRNAs contribute to the complex regulatory mechanisms that control HOXA gene expression.

I find the evolutionary analyses convincing; there is no doubt that these lncRNAs are ancient and that they share similar structures (though not necessarily sequence conservation) in vertebrates.

I am less convinced by the loss-of-function and gain-of-function experiments and by the associated transcriptomics analyses. Here are my main comments:

1) As the authors themselves acknowledge in this manuscript, the transcriptional organization of the HOX clusters is very complex. There are many alternative isoforms, on the sense or antisense strand, not to mention numerous regulatory elements embedded in the locus. The molecular consequences of gene editing in these loci are thus hard to predict, and have to be carefully analyzed before drawing a definitive conclusion. I thus wonder what exactly happens with the transcriptional organization of the locus in the CRISPRi and CRISPRa experiments. Are the preferred transcript start sites identical? Albeit in a different experimental setting (a targeted genomic deletion of the HOTAIR lncRNA), it was shown that the HOTAIR alteration leads to the emergence of a new lncRNA at the locus and potentially to transcriptional leakage on neighboring HOX genes (Amandio et al, PLoS Genetics, 2016). As unfortunately we still know little of the molecular consequences of CRISPRi and CRISPRa experiments, it is worth examining the resulting transcript organization with RNA-seq assays rather than examining a few target genes with qRT-PCR.

2) The authors do present an RNA-seq analysis for the siRNA knockdown performed on HT-29 cells (S2 table, qRT-PCR experiments in Figure 4). However, the differential expression analysis is not convincing: unless I have misunderstood the table (a detailed legend would help), there is no significant difference in expression for any HOX genes upon siRNA down-regulation of HOXA-AS3 and HOXB-AS3. In fact even the HOXA-AS3 and HOXB-AS3 transcripts do not show any significant expression change - I am not sure if this is because the expression levels are too low or too variable. This RNA-seq analysis (for the HOXA6-7 and HOXB6-7 genes) is not consistent with the qRT-PCR analysis shown in Figure 4. In general, qRT-PCR analyses, which show only fold changes but do not inform on the basal expression levels of the focus genes, ares not sufficient to prove that there is an effect on the target gene expression.

Overall, I think it is important to emphasize that all the experiments performed in the manuscript are performed in vitro, often on cancer cell lines that likely have numerous chromosomal alterations, and may not adequately reflect the situation in vivo. I would thus recommend that the results be interpreted with extreme caution.

**Have all data underlying the figures and results presented in the manuscript been provided?**

Reviewer #1: Yes

Reviewer #3: **No: **

Reviewer #4: Yes

PLOS authors have the option to publish the peer review history of their article (what does this mean?). If published, this will include your full peer review and any attached files.

Reviewer #1: No

Reviewer #3: **Yes: **Edwina McGlinn

Reviewer #4: No

---

## [Editor Report · Decision Letter 2]

24 Jun 2021

Dear Dr Ulitsky,

We are pleased to inform you that your manuscript entitled "Highly conserved and cis-acting lncRNAs produced from paralogous regions in the center of HOXA and HOXB clusters in the endoderm lineage" has been editorially accepted for publication in PLOS Genetics. Congratulations!

Yours sincerely,

Eric A. Miska

Associate Editor

PLOS Genetics

Bret Payseur

Section Editor: Evolution

PLOS Genetics

Comments from the reviewers (if applicable):

**Data Deposition**

http://datadryad.org/submit?journalID=pgenetics&manu=PGENETICS-D-20-01741R2

**Press Queries**

---

## [Editor Report · Acceptance letter]

6 Jul 2021

PGENETICS-D-20-01741R2 

Highly conserved and cis-acting lncRNAs produced from paralogous regions in the center of HOXA and HOXB clusters in the endoderm lineage 

Dear Dr Ulitsky, 

We are pleased to inform you that your manuscript entitled "Highly conserved and cis-acting lncRNAs produced from paralogous regions in the center of HOXA and HOXB clusters in the endoderm lineage" has been formally accepted for publication in PLOS Genetics! Your manuscript is now with our production department and you will be notified of the publication date in due course.

With kind regards,

Zsofi Zombor

PLOS Genetics

On behalf of:
